SciPost Physics

Submission

# 2D Fractons from Gauging Exponential Symmetries

Guilherme Delfino[1,&] and Claudio Chamon[1,*] and Yizhi You[2,#]

[1] *Department of Physics, Boston University, MA, 02215, USA*
[2] *Department of Physics, Northeastern University, MA, 02115, USA*

& delfino@bu.edu
* chamon@bu.edu
# y.you@northeastern.edu

August 11, 2023

## Abstract

The scope of quantum field theory is extended by introducing a broader class of discrete gauge theories with fracton behavior in 2+1D. We consider translation invariant systems that carry special charge conservation laws, which we refer to as exponential polynomial symmetries. Upon gauging these symmetries, the resulting $\mathbb{Z}_N$ gauge theories exhibit fractonic physics, including constrained mobility of quasiparticles and UV dependence of the ground state degeneracy. For appropriate values of theory parameters, we find a family of models whose excitations, albeit being deconfined, can only move in the form of bound states rather than isolated monopoles. For concreteness, we study in detail the low-energy physics and topological sectors of a particular model through a universal protocol, developed for determining the holonomies of a given theory. We find that a single excitation, isolated in a region of characteristic size $R$, can only move from its original position through the action of operators with support on $\mathcal{O}(R)$ sites. Furthermore, we propose a Chern-Simons variant of these gauge theories, yielding non-CSS type stabilizer codes, and propose the exploration of exponentially symmetric subsystem SPTs and fracton codes in 3+1D.

# 1  Introduction

After several decades of progress and development, the concept of topological order [1–3] has become a widely accepted framework for describing a diverse range of quantum states of matter. Quantum stabilizer codes have significantly contributed to the understanding of topological order through exactly-solvable Hamiltonians. Kitaev's toric code [4] and Wen's plaquette model [5] are prime examples of stabilizer Hamiltonians that effectively capture the essence of the underlying physics and describe an archetype of discrete $\mathbb{Z}_N$ gauge theories. Specifically, these models feature topological quasiparticles and ground state degeneracy derived from an exactly-solvable many-body spectrum and contribute to the understanding of topological entanglement entropy [6, 7], braiding statistics, and holonomies.

In this paper, we try to extend the scope of topological field theories by introducing a broader class of discrete gauge theories with fracton behavior in (2+1)D [8–11, 11, 12, 12–17]. For this, we consider quantum many-body systems that obey unusual charge conservation laws

$$G[f,g] = \sum_r g_r \, a^{f_r} \, q_r, \tag{1.1}$$

that we refer to as *exponential polynomial charges*. In the above, $G[f,g]$ are generators of continuum spatially modulated symmetries [18–21], where $q_r$ is the charge density at lattice point $r$, the integer-valued functions $f_r$ and $g_r$ have polynomial dependence on the lattice coordinates $r = (x,y)$, and $a$ is an integer parameter mod $N$. Our proposal is that the $\mathbb{Z}_N$ Higgsed phase of gauge theories coming from Eq. (1.1) might possess topological fracton order in two spatial dimensions, under some conditions on the theory parameters. As an aside, it is worth mentioning that for finite lattices, the cases we are interested in, $G[f,g]$ are always well-defined quantities and some care must be taken when periodic boundary conditions are imposed.

By gauging the exponential symmetries, the resultant gauge theory resembles the toric code model enriched with peculiar properties [22], such as fracton dynamics and a UV dependence of the ground state degeneracy on a torus [11–16, 16, 22–29, 29–32]. For several values of the theory parameters, we find resulting systems lacking string operators able to transport isolated unit charges and fluxes. Namely, the excitations in these theories, albeit deconfined, must move in the form of a bound state rather than a single unit, or be harnessed by creating additional charge excitations. This mobility restriction resembles

fracton physics in three dimensions, where isolated particles cannot move but their bound states can, with some important differences.

One of the many reasons fracton physics can be interesting is the presence of UV/IR mixing, where low-energy physics can depend on the lattice details. This is interesting, as appears to go against the principles of topological quantum field theories, where low-energy physics emerges from topologically robust entanglement patterns and is often immune to UV properties [6, 33]. Although fractonic systems are quite sensitive to UV details, their ground state manifolds are considered *topological*, in the sense that the ground state degeneracy is quite robust against arbitrary perturbations, and non-local operators are required to distinguish different ground states.

The models in this work are fractonic in the sense that they are all gapped, present long-range entanglement, and possess excitations that have their mobility constrained. Such models are slightly different from usual 3D fractons, though. While the latter usually presents a ground state degeneracy (GSD) that grows sub-extensively with the linear system size $L$ under periodic boundary conditions, our models display GSD that oscillates with system size but it does not scale sub-extensively. The degeneracies we find are consistent with Haah's upper bound on the GSD for homogeneous topological order that, in 2+1D, asserts that the GSD cannot scale with $L$ [34]. We remark that an extensive dependence of the GSD on $L$ needs not be a defining characteristic of fracton systems; sensitivity to the system size should suffice.

Chronologically, one of the first approaches to describe 3+1D topological fracton models beyond spin lattice Hamiltonians was through $\mathbb{Z}_N$ Higgsed phases of higher-multipole moment gauge theories [35, 36]. As one tries to do the same in 2+1D, however, a string of size $\mathcal{O}(N)$ emerges, allowing particles to move [13, 14, 16, 25, 37–39]. Later, people also figured that a possible mechanism for 3D fracton theories is subsystem symmetries [23, 40–42], where one has conserved quantities along sub-extensively many sub-manifolds. This mechanism does not seem to be effective for constructing 2D fracton topological systems, as gauging a subsystem symmetry in 2D does not result in local gauge fluctuations. In the meantime, the $\mathbb{Z}_N$ charge multipole symmetry [14] simplifies to sublattice symmetries and the resulting gauge theory can be characterized within the usual topological order framework. Here, we propose exponential symmetries as a possible mechanism to circumvent such challenges, where even under the Higgsing of $U(1)$ down to $\mathbb{Z}_N$ subgroup, no $\mathcal{O}(N)$ string operator emerge.

Our studies of exponential gauge symmetries also connect to the class of stabilizer code Hamiltonians studied in Ref. [22], if we fix $g_r = 1$ and $f_r = x + y$ in Eq. (1.1). In their work, the authors generalize Kitaev's $\mathbb{Z}_N$ toric code by introducing an integer parameter $a$. These models exhibit either topologically ordered or trivial phases depending on the specific parameter value $a$. Notably, for certain choices of $a$, such as $a = \mathrm{rad}(N)$, the system is no longer topologically ordered and presents no ground state degeneracy. We argue that in such cases, the generator in Eq. (1.1) can be viewed as local symmetry operators that only act on a finite number of degrees of freedom in the thermodynamic limit, implying that further gauging the exponential symmetry is not possible.

Although some works corroborate the inexistence of translation invariant topologically ordered fractonic systems in 2+1D [43] from discrete gauge theories, the models studied in this work provide a different perspective on the matter. In the following, we try to make more clear the fractonic behaviors we observe and provide an intuitive picture for their origin. Consider a single charge $q_0$ located at $r_0$, isolated from any other excitation in a large disk region of radius $R$ as depicted in Fig. 1. For an appropriate set of theory parameters $a$ and $N$, the excitation $q_0$ as well as its lattice position $r_0$, are stable against arbitrary perturbations. Only operators with support in $\mathcal{O}(R)$ sites are able to move $q_0$

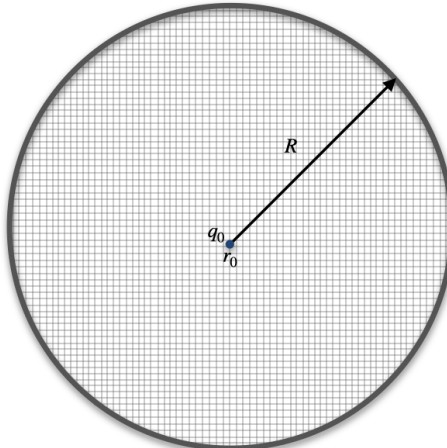

Figure 1: Under appropriate conditions on $a$ and $N$, an isolated charge $q_0$ cannot leave its lattice position $r_0$ unless non-local operators with support on $\mathcal{O}(R)$ sites act on it.

from $r_0$ to another site $r'$ inside the disk. Additionally, one has to consider $\mathcal{O}(R)$-th order terms in perturbation theory to see the tunneling process. The intuition is that in order for $G = \sum_r g_r \, a^{fr} \, q_r = q_0 \, g_{r_0} \, a^{f_{r_0}} + \sum_{|r|>R} g_r \, a^{fr} \, q_r$ to be conserved, the isolated excitation is allowed to move to a new position $r'$ inside the disk only if

$$q_0 \, g_{r_0} \, a^{f_{r_0}} = q_0 \, g_{r'} \, a^{f_{r'}} \mod N, \tag{1.2}$$

if nothing changes outside the disk. The mobility condition admits solutions or not depending on the theory parameters $N$ and $a$. For the case in which Eq. (1.2) admits no solutions, then the particle is constrained to not move at all, and isolated excitations behave as fractons - stable topological quasiparticles that cannot move and possess, as we later argue, anyonic statistics. Applying a similar analysis to bound states of two or more quasiparticles leads to the conclusion that these are able to move, characterizing the exponential symmetry fracton theories as Type-I fracton models.

Finally, the paper is organized as follows. In Section 2, we propose generalized 2+1D gauge theories that incorporate exponential charge symmetries and discuss how they can emerge from gauging bosonic lattice models. We also discuss how this construction is related to the models studied in Ref. [22]. In Section 3 we study, in detail, the low-energy physics of a concrete $\mathbb{Z}_N$ gauge theory coming from exponential charge symmetries. We discuss that, under some conditions over the parameters $a$ and $N$, the resulting theory corresponds to a Type-I fracton model. Finally, in Section 4 we discuss how the exponential symmetry gauging protocol can be used to build Chern-Simons-like theories and the interplay of exponential charge with sub-system symmetries in SPTs and 3D fracton phases.

## 2 2D Discrete Gauge Theory with Exponential Charge Symmetry

As a warm-up, let us begin by proposing a generalized electromagnetism framework that incorporates exponential charge conservation. We then demonstrate that this gauge theory provides an exact characterization of the Fuji-Cheng-Watanabe stabilizer code introduced in Ref. [22].

## 2.1 Gauging Exponential Charge Symmetry

To analyze the charge pattern in an electromagnetic theory on a 2D square lattice, we define a two-component 'electric field', denoted by $E_{1,\ell}$ and $E_{2,\ell}$, residing on the $x$- and $y$-links $\ell$ of the lattice. These fields are subject to a special Gauss law, parameterized by two integer parameters, $a_1$ and $a_2$. The charge $q_r$ at site $r$ of a square lattice

$$q_r = E_{1,r-\frac{e_x}{2}} - a_1 E_{1,r+\frac{e_x}{2}} + E_{2,r-\frac{e_y}{2}} - a_2 E_{2,r+\frac{e_y}{2}} \tag{2.1}$$

At this point, we assume that the system carries a $U(1)$ charge for generality. However, we will focus on the case of discrete charges in our later discussions. In the above, $e_x$ and $e_y$ are the lattice unit vectors along the $x$ and $y$ directions. Eq. (2.1) can be viewed as a generalized Gauss-law that incorporates exponential charge conservation, subject to appropriate boundary conditions. Thus, although the total charge density $\sum_q q_r$ is not conserved, the theory respects an *exponential charge conservation* [18,19,21,22], where

$$G = \sum_r a_1^x a_2^y q_r \tag{2.2}$$

is conserved. This is a special instance of the exponential polynomial generators $G[f,g]$ in Eq. (1.1) with $g_r = 1$ and $f_r = x + y$. This system can be also thought as a scalar charge theory where each particle located at position $(x,y)$ carries a site-dependent charge of $a_1^x a_2^y q_r$. As a result, the charge sectors undergo nontrivial transformations under translation operations.

One might be concerned about whether the quantity in Eq. (2.2) is well defined. In fact, for infinite systems $G$ diverges if $a_1, a_2 > 1$. In this work, we will be mostly interested in finite lattice systems with periodic boundary conditions, where $G$ is well-defined and does not diverge. Also, in the classes of systems we study in this work the charge in $G$ is conserved only modulo $N$, as we Higgs the $U(1)$ group down to $\mathbb{Z}_N$.

One possible way to describe the charge dynamics on the lattice, subject to the aforementioned special conservation law in Eq. (2.2), is through the bosonic model

$$b_r^{a_1} b_{r+e_x}^\dagger + b_r^{a_2} b_{r+e_y}^\dagger + \text{h.c.}. \tag{2.3}$$

Here, $b_r^\dagger$ and $b_r$ represent the boson creation and annihilation operators at position $r$ in the second-quantization language. Such charge dynamics lead to the creation of $a_1$ ($a_2$) charges at site $r$ and the annihilation of a unit charge at the neighboring $r + e_x$ ($r + e_y$) site. It is worth noting that the hopping term in Eq. (2.3) does not preserve global charge conservation, but it does respect the exponential charge conservation law in Eq. (2.2).

To clarify the gauge structure associated with the exponential symmetry described in Eq. (2.2), we introduce a coupling between the charged boson field and a gauge potential, denoted by $A_{1,\ell}$ and $A_{2,\ell}$

$$b_r^{a_1} e^{iA_{1,r+\frac{e_x}{2}}} b_{r+e_x}^\dagger + b_r^{a_2} e^{iA_{2,r+\frac{e_y}{2}}} b_{r+e_y}^\dagger + \text{h.c.}. \tag{2.4}$$

The gauge potential serves as the conjugate partner of the electric field and resides on the $x$- and $y$-links $\ell$ of the square lattice, respectively, illustrated in Fig. 2. The potential is subject to gauge transformations of the form

$$\begin{aligned} A_1 &\to A_1 - f_{r+e_x} + a_1 f_r, \\ A_2 &\to A_2 - f_{r+e_y} + a_2 f_r, \end{aligned} \tag{2.5}$$

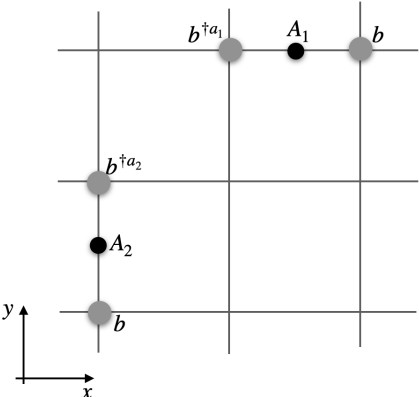

Figure 2: The charged bosonic fields $b$ live at the vertices of the square lattice, while the gauge fields $A_1$ and $A_2$ are defined at the horizontal and vertical edges.

which explicitly depends on the theory parameters $a_1$ and $a_2$.

We hereby define the leading order, gauge-invariant, magnetic flux operator,

$$B_{\tilde{r}} = A_{1,\tilde{r}+\frac{e_y}{2}} - a_2 A_{1,\tilde{r}-\frac{e_y}{2}} - A_{2,\tilde{r}+\frac{e_x}{2}} + a_1 A_{2,\tilde{r}-\frac{e_x}{2}}, \qquad (2.6)$$

which lives at the center of the plaquette on the dual lattice $\tilde{r}$ illustrated as Fig. 3. To simplify the notation, we introduce two operators $D_i$ and $\tilde{D}_i$ that can be viewed

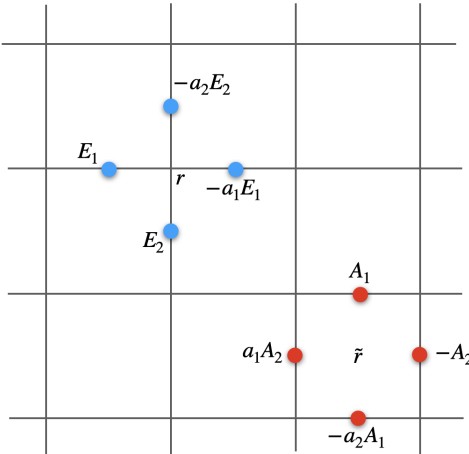

Figure 3: Charge conservation law at $r$ and magnetic flux operators at $\tilde{r}$ defined on the square lattice.

as generalized differential polynomials operators defined on the original and dual square lattice, respectively

$$
\begin{aligned}
D_1 f_r &= -a_1 f_{r+\frac{e_x}{2}} + f_{r-\frac{e_x}{2}} \\
D_2 f_r &= -a_2 f_{r+\frac{e_y}{2}} + f_{r-\frac{e_y}{2}} \\
\tilde{D}_1 f_{\tilde{r}} &= -f_{\tilde{r}+\frac{e_x}{2}} + a_1 f_{\tilde{r}-\frac{e_x}{2}} \\
\tilde{D}_2 f_{\tilde{r}} &= -f_{\tilde{r}+\frac{e_y}{2}} + a_2 f_{\tilde{r}-\frac{e_y}{2}}
\end{aligned}
\qquad (2.7)
$$

Using this notation, we can express the Gauss law and magnetic flux operators in the

compact form

$$\begin{aligned} D_1 E_1 + D_2 E_2 &= q, \\ \tilde{D}_1 A_2 - \tilde{D}_2 A_1 &= B, \end{aligned} \tag{2.8}$$

where we stopped explicitly writing the dependence on the lattice points for convenience.

## 2.2 Discretization to $\mathbb{Z}_N$ Charge

We establish a connection between our theory and the Fuji-Cheng-Watanabe stabilizer code in Ref. [22]. We discretize our gauge theory by adding a $\cos(N q_r)$ term to the Hamiltonian, which energetically favors $\mathbb{Z}_N$ charges in the low-energy states of the theory. As a result, the corresponding electromagnetic fields can be expressed in terms of the $\mathbb{Z}_N$ Pauli operators $\omega^E = X$ and $e^{iA} = Z$, with $\omega = e^{2\pi i/N}$. The canonical quantization between the vector potential and the electric field components recovers the $\mathbb{Z}_N$ Pauli algebra $XZ = \omega ZX$ for operators at the same edges. It is worth noting that the resulting gauging theory is equivalent to the stabilizer code in Ref. [22], as

$$H = -\sum_r Q_r - \sum_{\tilde{r}} B_{\tilde{r}} + \text{h.c.}, \tag{2.9}$$

where

$$\begin{aligned} Q_r &= X_{r+\frac{e_x}{2}}^{-a_1} X_{r-\frac{e_x}{2}} X_{r+\frac{e_y}{2}}^{-a_2} X_{r-\frac{e_y}{2}}, \\ B_{\tilde{r}} &= Z_{\tilde{r}+\frac{e_x}{2}}^{-1} Z_{\tilde{r}-\frac{e_x}{2}}^{a_1} Z_{\tilde{r}+\frac{e_y}{2}} Z_{\tilde{r}-\frac{e_x}{2}}^{-a_2}, \end{aligned} \tag{2.10}$$

as shown in Fig. 4. The ground states of the above Hamiltonian are projected onto the vanishing charge and flux sector by the $Q_r$ and $B_r$ operators. As discussed in Ref. [22], the model realizes either topologically ordered phases or trivial phases depending on the parameters $a_1, a_2$ in the model.

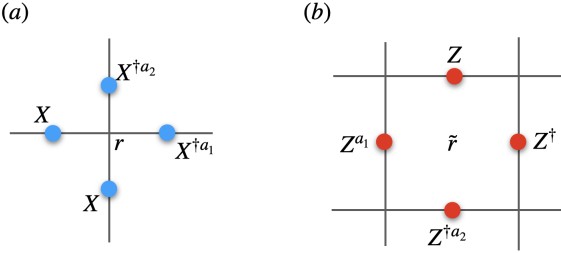

Figure 4: a) Operators for the $\mathbb{Z}_N$ a) charge $Q_r$ and b) flux $B_{\tilde{r}}$ at sites $r$ and $\tilde{r}$ at original and dual lattices, respectively.

First, we will comment on the case where $a_i = 0 \mod \text{rad}(N)$, i.e., when $a_i$ are integer multiples of $\text{rad}(N)$. The ground state manifold of the model in such case corresponds to a trivial phase that can be disentangled using finite-depth local unitary circuits, as discussed in Ref. [22]. In our exponential charge symmetry construction, one can understand this coming from the fact that when gauging the matter field in Eq. (2.3), it is essential to be cautious if the charge is a discrete $\mathbb{Z}_N$ field. If $a_i \neq 0 \mod \text{rad}(N)$, the exponential symmetry operator $G$ defined in Eq. (2.2) is a global symmetry, and the gauging process can proceed smoothly, as we have proposed. However, when $a_i = p \, \text{rad}(N)$, for some $p \in \mathbb{Z}$, we can infer that there exists an integer $m$ such that $a_i^m = 0 \mod N$. Consequently, the

exponential symmetry only acts on a finite number of degrees of freedom for sites $|r| < m$ as $G$ becomes a local symmetry in the thermodynamic limit, after coarse-graining. Since a local symmetry cannot be further gauged, we do not expect it giving rise to any additional gauge structure. This is why the stabilizer code in Ref. [22] exhibits a trivial phase when $a_i = \text{rad}(N)$.

To simplify further discussions, without loss of generality, we consider $a_1 = a_2 \equiv a$ and fix the system size to $L$. To illustrate the effects of the $\mathbb{Z}_N$ exponential symmetry under boundary conditions, we express the $U(1)$ symmetry generator in Eq. (2.2) in terms of the $\mathbb{Z}_N$ charge operators in Eq. (2.10) by an exponential map $G_N \sim e^G$

$$G_N = \prod_r Q_r^{a^x + a^y}. \tag{2.11}$$

Due to periodic boundary conditions, it is required that under $L$ sites translation (either in $x$ or $y$ directions)$G_N$ must be a well defined quantity, that is, $a^L = a^0 \mod N$. When $a^L - 1 = 0 \mod N$, the exponential symmetry is well-defined on a closed manifold, and the resulting theory is reminiscent of a $\mathbb{Z}_N$ gauge theory. However, when $a^L - 1 \neq 0 \mod N$, the $\mathbb{Z}_N$ exponential symmetry must be reduced to a $\mathbb{Z}_k$ symmetry,

$$G_k = \prod_r Q_r^{\frac{N}{k}(a^x + a^y)}, \tag{2.12}$$

with $k = \gcd(a^L - 1, N)$ in order to be compatible with the periodic boundary conditions. In the previous expression, $\gcd(A, B)$ stands for the greatest common divisor between the two positive integer numbers $A$ and $B$. The resulting gauge theory should be akin to a $\mathbb{Z}_k$ gauge theory, explaining the size dependence of the ground state degeneracy observed in Ref. [22]. Here we mention that the result $k = \gcd(a^L - 1, N)$ holds only for $a$ and $N$ coprime. We discuss the more general case in Appendix 5.1 in the context of the model proposed in Section 3 which similarly holds for this system.

In the following section, we study a particular instance of gauge theories coming from exponential charge symmetries. We argue that such theories can be quite constraining for the quasiparticles mobility and might host fracton-like excitations, under some conditions on their defining parameters.

# 3 Global and Exponential Charge Symmetries Gauge Theories

Our general goal is to establish a class of generalized discrete gauge theories in which charge and flux excitations undergo non-trivial transformations under translation operations. The Fuji-Cheng-Watanabe stabilizer code, perhaps the simplest model in this class, has already demonstrated its potential for revealing 'fractonic' and post-RG critical phenomena. In general, the exponential symmetries generated by $G[f, g]$ in Eq. (1.1) do not commute with translations and discrete rotations and, therefore, cannot be regarded as internal symmetries. A direct consequence of this is the coupling between IR physics and UV regularization, where the ground state degeneracy depends on the lattice details. Meanwhile, we anticipate that anyon excitations in the associated gauge theory undergo nontrivial permutations in response to lattice translations and lattice defects.

For concreteness, we now focus on a special instance of exponential polynomial symmetries in Eq. (1.1) that conserves both usual global charge as well as exponential charge. As we are going to argue, this combination appears to be enough for the emergence of

topologically robust quasiparticles with restricted mobility. In practice, we fix the polynomial functions of lattice points $r = (x, y)$ to be $g_r = 1$ and allow $f_r$ to run over the choices of 1, $x$, $y$, and $x+y$ on a square lattice. This engender four independent symmetry generators

$$G = \sum_r q_r, \qquad\qquad G_x = \sum_r a^x\, q_r,$$

$$G_y = \sum_r a^y\, q_r, \qquad\qquad G_{xy} = \sum_r a^{x+y}\, q_r. \tag{3.1}$$

The symmetry defined in Eq. (3.1) respects both usual global charge and exponential charge symmetry. Based on this special conservation law, the possible charge dynamics on the lattice can be written as

$$b_r^{\dagger a}\, b_{r+e_x}^{a+1}\, b_{r+2e_x}^{\dagger} + b_r^{\dagger a}\, b_{r+e_y}^{a+1}\, b_{r+2e_y}^{\dagger} + \text{h.c.}. \tag{3.2}$$

Although other terms in the bosonic Hamiltonian are allowed, such as longer-range hopping terms, they can be generated as higher-order combinations of the boson hopping term in Eq. (3.2). Therefore, we neglect these for now as they do not affect the gauge structure. We see that the price for imposing the symmetries generated by Eq. (3.1) is to consider collective coordinate hopings of bosons among three sites, in contrast to the usual two-sites kinetic terms.

The boson dynamics described in Eq. (3.2) can be interpreted as the creation of $a$ number of $x$-dipoles units (or $y$-dipoles) and the annihilation of a unit charge on the nearby $r + e_x$ (or $r + e_y$) site. Here, a dipole unit refers to a pair of charge and hole separated along the $x$ (or $y$) link. One can explicitly check that these hopping terms respect all the symmetries defined in Eq. (3.1), and consequently, terms like those in Eq. (2.3) are not allowed as they violate the total charge number $G$ conservation. In this sense, the exponential charge dynamics are generated at the dipole level.

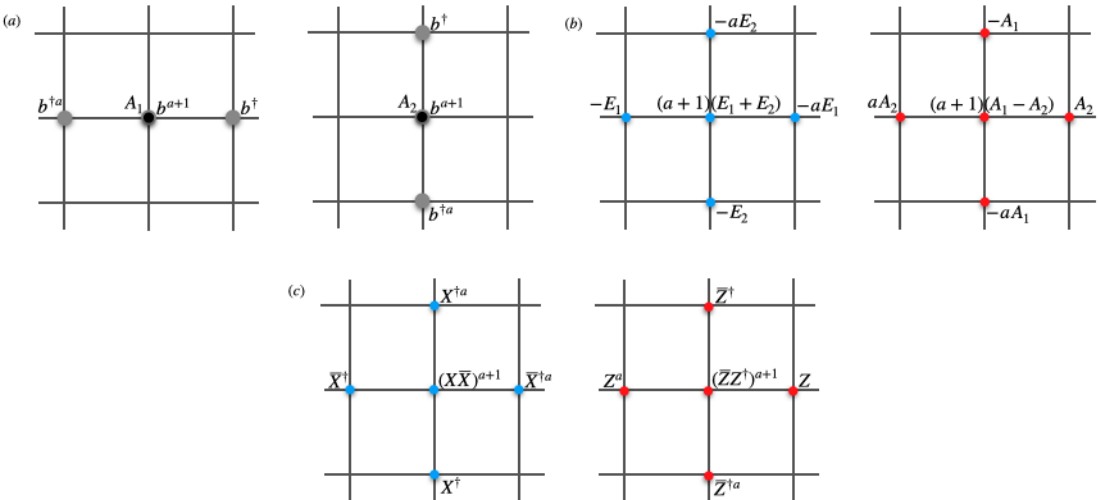

Figure 5: (a) Boson hopping terms, together to the gauge fields, respecting all conserved charges in Eq. (3.1); (b) Gauss Law and magnetic flux operator in the $U(1)$ gauge theory; (c) Charge and flux operators $\mathcal{Q}_r$ and $\mathcal{B}_r$, respectively, after Higgsing $U(1)$ down to $\mathbb{Z}_N$.

To elucidate the gauge structure, we gauge the symmetry in Eq. (3.1) by coupling the boson field with a gauge potential $A_1$, $A_2$, as illustrated in Fig. 5(a),

$$(b_r^{\dagger})^a\, e^{iA_{1,r+e_x}}\, b_{r+e_x}^{a+1}\, b_{r+2e_x}^{\dagger} + (b_r^{\dagger})^a\, e^{iA_{2,r+e_y}}\, b_{r+e_y}^{a+1}\, b_{r+2e_y}^{\dagger}. \tag{3.3}$$

The gauge fields $A_1$ and $A_2$ mediate the interaction among the bosons on the three sites, and consequently, live on the vertices of the square lattice, as well as their conjugate partners electric fields $E_1$ and $E_2$. This is to be contrasted to the example in the previous section, where the gauge fields lived on the lattice edges. The gauge potentials are subject to the following transformations

$$A_{1,r} \rightarrow A_{1,r} + f_{r+e_x} + a f_{r-e_x} - (a+1) f_r,$$
$$A_{2,r} \rightarrow A_{2,r} + f_{r+e_y} + a f_{r-e_y} - (a+1) f_r. \tag{3.4}$$

The generator of these gauge transformations is the genereralized Gauss law

$$D_1 E_1 + D_2 E_2 = q_r, \tag{3.5}$$

where we defined the "lattice derivative operators"

$$D_1 f_r = -f_{r+e_x} - a f_{r-e_x} + (a+1) f_r,$$
$$D_2 f_r = -f_{r+e_y} - a f_{r-e_y} + (a+1) f_r. \tag{3.6}$$

It is a straightforward exercise to show that with this Gauss law, the quantities in Eq. (3.1) are automatically conserved in an infinite lattice.

We define the gauge invariant magnetic flux operator, which also lives at the vertices of the square lattice,

$$B_r \equiv a A_{2,r-e_x} + A_{2,r+e_x} - (a+1) A_{2,r} - a A_{1,r-e_y} - A_{1,r+e_y} + (a+1) A_{1,r}, \tag{3.7}$$

as illustrated in Fig. 5(b). After some algebra, it can be deduced that the magnetic flux operator results in the following conservation laws, on an $L_x \times L_y$ periodic lattice

$$\sum_r B_r = 0, \qquad \sum_r a^{L_x - x} B_r = 0,$$
$$\sum_r a^{L_y - y} B_r = 0, \qquad \sum_r a^{L_x + L_y - x - y} B_r = 0. \tag{3.8}$$

The similarity with the charge conservations in Eq. (3.1) is no coincidence, as the charge and flux operators are dual to each other. As a result, both the total usual flux and the 'exponential flux' are conserved.

Again, we discretize our $U(1)$ gauge theory down to $\mathbb{Z}_N$, effectively implementing $\mathbb{Z}_N$ charges. Now, since the gauge fields are hosted on the vertices of the lattice, there will be two $\mathbb{Z}_N$ degrees of freedom per lattice site. We parameterize the fields components as $\omega^{E_2} = X$, $e^{iA_2} = Z$, $\omega^{E_1} = \bar{X}$, and $e^{iA_1} = \bar{Z}$. The resulting gauge theory can be written in terms of the CSS-type code,

$$H = -\sum_r \mathcal{Q}_r - \sum_r \mathcal{B}_r + \text{h.c.} \tag{3.9}$$

with $\mathcal{Q}_r$ and $\mathcal{B}_r$ the $\mathbb{Z}_N$ charge and flux operators, defined as

$$\mathcal{Q}_r = X_r^{a+1} \bar{X}_r^{a+1} \bar{X}_{r+e_x}^{\dagger a} \bar{X}_{r-e_x}^\dagger X_{r+e_y}^{\dagger a} X_{r-e_y}^\dagger,$$
$$\text{and} \quad \mathcal{B}_r = Z_r^{\dagger a+1} \bar{Z}_r^{a+1} \bar{Z}_{r+e_y}^\dagger \bar{Z}_{r-e_y}^{\dagger a} Z_{r+e_x} Z_{r-e_x}^a, \tag{3.10}$$

as illustrated in Fig. 5(c).

First, we emphasize that the aforementioned protocol only works when $a \neq 0$ mod $\text{rad}(N)$, so the symmetries we begin with are global symmetries. If $a = p\,\text{rad}(N)$, for some $p \in \mathbb{Z}$ one can infer that there exists a finite integer $m$ such that $a^m = 0$ mod $N$. Consequently, the exponential symmetry acts only on a finite number of degrees of freedom for sites $|r| < m$ such local symmetry cannot be further gauged. We now study the low-energy physics of the Hamiltonian in Eq. (3.9), such as its ground state properties, topological sectors, and elementary excitations.

## 3.1 Ground State Degeneracy

The phase described by the ground state of the Hamiltonian in Eq. (3.9) is gapped and topologically ordered. The model sits at an exactly solvable point, allowing us to solve the system exactly. We now count the ground state degeneracy when the system is put on a torus, which can be non-trivial on compact lattices. Every term in the Hamiltonian commute with each other so we can diagonalize all of them simultaneously. As each one of the operators obey $\mathcal{Q}_r^N = \mathbb{1}$ and $\mathcal{B}_r^N = \mathbb{1}$, they have eigenvalues $e^{2\pi i p/N}$ for $p = 0, 1, \ldots, N-1$. Here, we focus on the ground state space, corresponding to the eigenvalues $+1$, minimizing the Hamiltonian energy

$$\mathcal{H}_0 = \{|\psi\rangle \in \mathcal{H} \mid \mathcal{Q}_r |\psi\rangle = +1 |\psi\rangle \text{ and } \mathcal{B}_r |\psi\rangle = +1 |\psi\rangle\}. \quad (3.11)$$

The ground state space is characterized by the local requirement of vanishing $\mathbb{Z}_N$ charge and flux everywhere. In such case, one can distinguish elements in $\mathcal{H}_0$, if degenerate, only through global features, which is the essence of topologically ordered states.

In the following, we assume periodic boundary conditions for the lattice in both $x$ and $y$ directions. This introduces consistency requirements on the field boundary configurations such that the conserved quantities in Eq. (3.1) and (3.8) are well defined. In terms of the $\mathbb{Z}_N$ degrees of freedom, the conservation equations in Eq. (3.1) translate into

$$\prod_r \mathcal{Q}_r = \mathbb{1},$$
$$\prod_r \mathcal{Q}_r^{\rho_1 a^x} = \mathbb{1},$$
$$\prod_r \mathcal{Q}_r^{\rho_2 a^y} = \mathbb{1},$$
$$\prod_r \mathcal{Q}_r^{\rho_{12} a^{x+y}} = \mathbb{1}, \quad (3.12)$$

and similarly for the flux conservations in Eq. (3.8) in terms of the $\mathcal{B}_r$ operators. In the above, $N_a$ is the greatest divider of $N$ that is coprime to $a$, and the $\rho$ factors are given by

$$\rho_1 = \frac{N_a}{\gcd(N_a, a^{L_x} - 1)},$$
$$\rho_2 = \frac{N_a}{\gcd(N_a, a^{L_y} - 1)},$$
$$\rho_{12} = \frac{N_a}{\gcd(N_a, a^{L_x} - 1, a^{L_y} - 1)}, \quad (3.13)$$

where $L_x$ and $L_y$ are the linear sizes of the lattice in the $x$ and $y$ directions. For a detailed derivation of Eq. (3.12) and the $\rho$ factors, see Appendix 5.1.

The twisted periodic boundary conditions are reflected on the ground state degeneracy, which depends sensitively on the lattice sizes in a rather non-usual way

$$\dim \mathcal{H}_0 = \left[ N \gcd(N_a, a^{L_x} - 1) \gcd(N_a, a^{L_y} - 1) \gcd(N_a, a^{L_x} - 1, a^{L_y} - 1) \right]^2, \quad (3.14)$$

which we explain in detail in Appendix 5.1. This result has a functional dependency very different from the usual 3D fracton physics [44–47] and 2D higher multipole moment conserving models [13, 15, 48]. It presents a more exotic manifestation of UV/IR mixing, depending on factors as $a^L$, a direct result of the underlying exponential charge gauge structure. Finally, it is worth mentioning that this result is in agreement with the general upper bound on the ground state degeneracy for homogeneous topological ordered systems [34], as $\dim \mathcal{H}_0 \leq N^8$ and does not scale with the system size.

## 3.2 Wilson Loops and Holonomies

In gauge theories, holonomies from the Wilson line operators reveal the global flux sectors of the ground state manifold on a torus and determine the Wilson Algebra. In this section, we derive Wilson operators [51] for the model in Eq. (3.9) based on its gauge structure. For fracton gauge theories defined on a lattice, the Wilson operators are more complex compared to conventional TQFT because: i) Fracton dynamics and multipole charge conservation law make some Wilson lines unbendable and direction-specific; ii) There exist other higher-dimensional Wilson operators with unique shapes like membranes or cages, depending on the lattice geometry [29–31]; iii) Parallel Wilson operators might be inequivalent, with the possibility of a sub-extensive number of independent Wilson line operators.

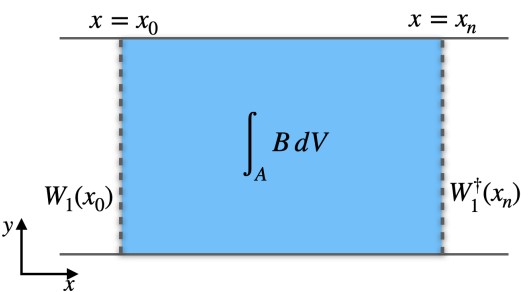

Figure 6: Total flux on an open area $\mathcal{A}$ (depicted in blue) reduces to string operators acting at the boundaries $x = x_0$ and $x = x_n$.

To obtain the Wilson algebra, we will apply the protocol developed in Ref. [51] to our model. In order to obtain the Wilson operators, we need to figure out the generalized Gauss Law and Stokes theorem that relate operators on a closed surface to the total charge/flux inside the surface. Thus, we revisit the magnetic and electric conservation laws in Eqs. (3.1) and (3.8) over a closed area as shown in Fig. 6. The magnetic conserved quantities provide us $y-$ oriented Wilson line operators crossing the system at coordinate $x$ (for further details of derivation, see Appendix 5.2) given by

$$
\begin{aligned}
W^{(1)}(x) &= \prod_i Z^a(x,i)\, Z^\dagger(x+1,i), \\
W^{(2)}(x) &= \prod_i Z(x,i)\, Z^\dagger(x+1,i), \\
W^{(3)}(x) &= \prod_i Z^{a^{L_y-i+1}}(x,i)\, Z^{\dagger a^{L_y-i}}(x+1,i), \\
W^{(4)}(x) &= \prod_i Z^{a^{L_y-i}}(x,i)\, Z^{\dagger a^{L_y-i}}(x+1,i).
\end{aligned}
\tag{3.15}
$$

The $x$-oriented Wilson loops $W_x^{(i)}(y)$ for $i = 1,\dots,4$, crossing the system at coordinate $y$, are also given by similar expressions (see Appendix 5.2). Here and in the following, the orientation of the string operators are implicit in their coordinate dependence $W(x)$ for $y-$oriented and $W(y)$ for $x-$oriented ones. The dual Wilson lines, associated with the

electric charge conservation laws in Eq. (3.1), are

$$
\begin{aligned}
V^{(1)}(y) &= \prod_i X^\dagger(i,y)\, X^a(i,y+1), \\
V^{(2)}(y) &= \prod_i X^\dagger(i,y)\, X(i,y+1), \\
V^{(3)}(y) &= \prod_i X^{\dagger a^i}(i,y)\, X^{a^{i+1}}(i,y+1), \\
V^{(4)}(y) &= \prod_i X^{\dagger a^i}(i,y)\, X^{a^i}(i,y+1),
\end{aligned} \tag{3.16}
$$

and, likewise for the $y$-oriented dual loops $V^{(i)}(x)$. It is worth mentioning that all the $W^{(i)}$ and $V^{(i)}$ operators have support on closed double parallel strings. Also, they are all symmetries of the Hamiltonian and form two copies of a Wilson line algebra, spanning degenerate states in Hilbert space that are locally indistinguishable. Not all $W^{(i)}$ and $V^{(i)}$ operators commute with each other, as they have a non-vanishing intersection. The non-trivial algebra between these operators and its action on the Hilbert space ensures the degeneracy of the ground state space (as well as a topological degeneracy in all other energy sectors).

For later purposes, it is useful to note that all the $W^{(i)}(x)$ and $V^{(i)}(y)$ operators can be constructed from the single-line uniform and exponentially weighted strings

$$
\begin{aligned}
F^{(1)}(x) = \prod_i Z(x,i), \qquad & F^{(2)}(x) = \prod_i Z(x,i)^{a^{L_y-i}}, \\
G^{(1)}(y) = \prod_i X(i,y), \qquad & G^{(2)}(y) = \prod_i X(i,y)^{a^i}.
\end{aligned} \tag{3.17}
$$

In order to specify the Wilson line algebra and recover the ground state degeneracy in Eq. (3.14), it is useful to choose specific combinations of the line operators in Eq. (3.17), which carry the same amount of information as $\{W^{(i)}, V^{(i)}\}$,

$$
\begin{aligned}
S^{(1)}(y) &\equiv G^{(1)}(y), \\
S^{(2)}(y) &\equiv G^{(1)}(y)\, G^{(1)\dagger}(y+1), \\
S^{(3)}(y) &\equiv G^{(2)}(y)\, G^{(1)\dagger}(y), \\
S^{(4)}(y) &\equiv S^{(3)}(y)\, S^{(3)\dagger}(y+1),
\end{aligned} \tag{3.18}
$$

and

$$
\begin{aligned}
T^{(1)}(x) &\equiv F^{(1)}(x), \\
T^{(2)}(x) &\equiv F^{(2)}(x)\, F^{(1)\dagger a^{L_y}}(x), \\
T^{(3)}(x) &\equiv F^{(1)}(x)\, F^{(1)\dagger}(x+1), \\
T^{(4)}(x) &\equiv T^{(2)}(x)\, T^{(2)\dagger}(x+1).
\end{aligned} \tag{3.19}
$$

The set of operators $\{S^{(i)}, T^{(i)}\}$ is useful as it spans and enumerates the different topological sectors of the ground state manifold. First, one can explicitly check that each one of the $S^{(k)}$ operators commute with all others but with their corresponding pair $T^{(k)}$, namely, $[S^{(i)}, T^{(j)}] = 0$ unless $i = j$. Now, we detail the algebra of each pair $(S^{(k)}, T^{(k)})$ for $k = 1, \ldots, 4$ and recover $\dim \mathcal{H}_0$ in Eq. (3.14).

For $k = 1$, $S^{(1)}(y)$ and $T^{(1)}(x)$ are $\mathbb{Z}_N$ operators regardless of the system size. It is not difficult to demonstrate that these operators follow the Wilson line algebra

$$S^{(1)}(y)T^{(1)}(x) = \omega T^{(1)}(x)S^{(1)}(y) \tag{3.20}$$

and, thus, $(S^{(1)}(y), T^{(1)}(x))$ span an $N$−fold degenerate Hilbert space.

For the rest of operator pairs, one needs extra caution when evaluating their eigenvalues. For $k = 2$, $T^{(2)}(x)$ reduces to a $\mathbb{Z}_{\gcd(a^{L_y}-1,N)}$ operator under closed boundary conditions, so its eigenvalues can only change mod $\gcd(a^{L_y} - 1, N)$. Likewise, while $S^{(2)}(y)$ is well-defined regardless of system size, it obeys the extra constraint $S^{(2)}(y) \, S^{(2)\dagger a_y^L}(y+L_y) = 1$. This again, confirms that the $S^{(2)}(y)$ eigenvalues can only change mod $\gcd(a^{L_y} - 1, N)$. Thus, $(S^{(2)}(y), T^{(2)}(x))$ span a Hilbert space that is $\gcd(a^{L_y} - 1, N)$-fold degenerate. Following the same argument for $k = 3$, $(S^{(3)}(y), T^{(3)}(x))$ also span a $\gcd(a^{L_x} - 1, N)$-fold degenerate Hilbert space.

Finally, for $k = 4$, $S^{(4)}(y)$ reduces to a $Z_{\gcd(a^{L_y}-1,N)}$ operator under closed boundary conditions and its eigenvalues can only change mod $\gcd(a^{L_y} - 1, N)$. Additionally, the constraint $T^{(4)\dagger a_x^L}(x)T^{(4)}(x + L_x) = 1$ reduces the eigenvalues of $T^{(4)}(x)$ to elements of $\mathbb{Z}_{\gcd(a^{L_y}-1,a^{L_x}-1,N)}$. Thus, $(S^{(4)}(y), T^{(4)}(x))$ span a $\gcd(a^{L_y} - 1, a^{L_x} - 1, N)$−fold degenerate Hilbert space. Putting all these degeneracy factors together, we are able to reconstruct the GSD expression in Eq. (3.14).

Since closed strings are all symmetries of the Hamiltonian, these operators allow us to navigate across the ground state manifold. The fact that all such operators are non-local, with support in $\mathcal{O}(L_x, L_y)$ sites, ensures that the ground states are topologically ordered, and all excitations created at the endpoints of open strings are quite stable against perturbations.

## 3.3 Fractionalized Excitations

We now turn to the study of excited states in the spectrum of the Hamiltonian $H$ in Eq. (3.9). Due to the fixed-point nature of $H$, all dispersion bands are flat and excitations are localized in real space. Charge and flux quasiparticles are associated with eigenvalues of $\mathcal{Q}_r$ and $\mathcal{B}r$ that differ from $+1$. As in usual gauge theories, these excitations emerge at the endpoints of open strings or higher-dimensional objects. Here, we study open versions of the double-string holonomies $W^{(i)}$ and $V^{(i)}$. From inspection, one can show that all these operators create four-particles bound states at each of their endpoints that are fully mobile on the lattice, ensuring they move collectively. For illustration, in Fig. 7, we show the excitation pattern created by open string operators $W_y^{(1)}$ and $W_y^{(2)}$, of size $s$ at coordinate $y = y_0$, when acted on a ground state.

As illustrated in Fig. 7, the role of $W^{(i)}$ (and also $V^{(i)}$) is to create quadrupole bound states at their endpoints. The excitations, when studied individually, can possess quantum numbers that are position-dependent, suggesting that isolated excitations or dipole-bound states can have constrained mobility. As we argue in the next section, in fact, dipolar bound states and isolated excitations mobility are rather restricted, and correspond to lineons and fractons, respectively.

## 3.4 Excitations with Constrained Mobility

While open $W^{(i)}$ and $V^{(i)}$ string operators create fully mobile excitations, here we argue that open $F^{(i)}$ and $G^{(i)}$ strings create, at their endpoints, dipolar bound states that are lineons, i.e., that can only move along 1-dimensional sublattices. Fig. 8 illustrates the excitation creation pattern at the endpoints of such strings. The restriction on the dipole

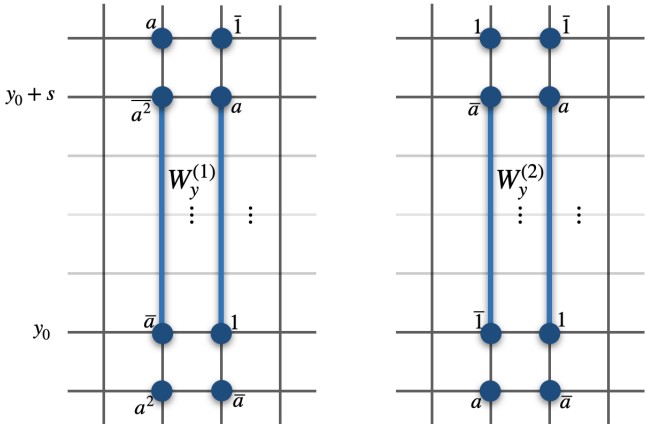

Figure 7: Open $W_y^{(i)}$ operators, for $i = 1$ and 2, create four quasiparticles bound states of excitations at their endpoints.

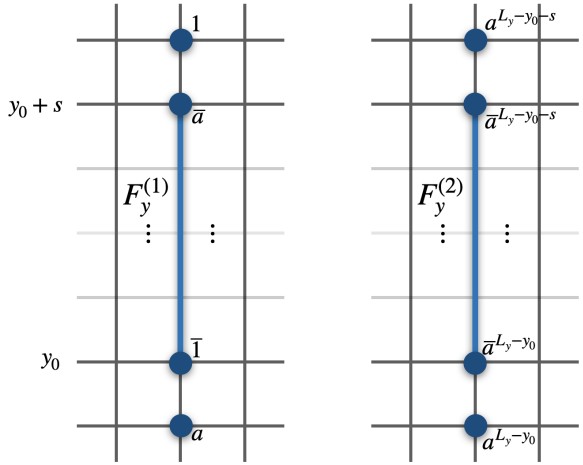

Figure 8: Open $F_y^{(1)}$ and $F_y^{(2)}$ operators create two quasiparticles bound states of excitations at their endpoints. The quantum numbers of excitations depend explicitly on the lattice position.

mobility derives from the fact that one cannot bend their corresponding lines without exciting additional quasiparticles from the vacuum. All one can do to move dipole-bound states is to stretch or compress $F^{(i)}$ and $G^{(i)}$ along straight lines, indicating that such bound states correspond to lineons.

We can also consider operators that are able to move isolated excitations, that is, open string operators that only violate a single $\mathcal{Q}_r$ (or $\mathcal{B}_r$) terms at their endpoints. It turns out that such operators introduce a length scale in the system - the minimal step with which an isolated particle can move. Let us define the following sum of integer powers of $a$

$$\alpha_i = \sum_{j=0}^{i} a^j = \frac{a^{i+1} - 1}{a - 1} \mod N, \qquad (3.21)$$

which can assume integer values mod $N$. Strings of Pauli operators weighted by $\alpha_i$ func-

tions

$$M_{y_0,s}(x) = \prod_{i=1}^{s} Z^{\alpha_{s-i}}(x, y_0 + i),$$

$$N_{x_0,s}(y) = \prod_{i=1}^{s} X^{\alpha_i}(x_0 + i, y), \tag{3.22}$$

are responsible for creating isolated excitations at one of their endpoints, as shown in Fig. 9. As illustrated in the figure, at the other endpoint, however, two excitations are created.

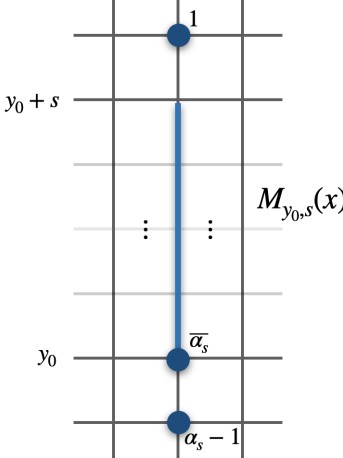

Figure 9: Open $M_{y_0,s}(x)$ operators create one isolated excitation at one of its end-points and two others at the opposite one.

This fact implies that the commensurability among $a$ and $N$ might introduce a length scale in the system. To be more precise, consider $\ell$ to be the smallest integer such that

$$\alpha_\ell = 1 \mod N. \tag{3.23}$$

This is the mobility condition for this theory, analogous to Eq. (1.2). When the Eq. (3.23) has a solution, we can fix $s = \ell$ and the excitation at $(x, y_0 - 1)$, as shown in Fig. 9, condenses back to the vacuum. In this case, effectively, $M_{y_0,s}(x)$ has the role of creating a $-1$ and $+1$ charges at $(x, y_0)$ and $(x, y_0 + \ell + 1)$, respectively. Thus, isolated particles of unit charge can hop with step sizes $\ell$ in both $x$ and $y$ directions and the resulting theory is not intrinsically fractonic. It is a straightforward exercise to check that for $a = 1$, $\alpha_i = i + 1$ and the smallest solution to $\alpha_\ell = 1$ is $\ell = N$. For $a = 1$ we recover a variation of the $\mathbb{Z}_N$ code studied in Ref. [25], where the emergence of such $N$-sized string operators plays an important role.

We are interested, however, in the class of theories where Eq. (3.23) does not admit solutions, which might happen when $a$ and $N$ share common factors in their prime decomposition. In such cases, there are no string operators in the theory able to move a single excitation, engendering an effective two-dimensional fracton system. Naively one would expect, as in usual 3D fracton theories, an extensive number of nonequivalent anyons in the theory, one for each lattice point, since they are not connected by string operators. This is not the case, however, as the number of independent anyons is captured by the ground state degeneracy on a torus (Eq. (3.14)) and as we argued before, is $\mathcal{O}(1)$ in $L$. An intuition for this comes from the fact that closed versions of the strings in Eq. (3.22) are

not symmetries of the Hamiltonian. In other words, they cannot connect degenerate states in different topological sectors since they are not holonomies of the theory. In contrast, the holonomies $\{W^{(i)}V^{(i)}\}$ and $\{S^{(i)}T^{(i)}\}$, and even the $\{F^{(i)}VG(i)\}$ operators, are the ones responsible for spanning the degenerate topological Hilbert space sectors.

It is interesting to note that Eq. (3.23), for general $N$ and $a \neq 0 \mod \text{rad}(N)$, is a variation of the order-finding problem, a question which is known for being a hard problem to solve in classical computers for large enough $a$ and $N$. To see the connection, we use the closed form for $\alpha_i$ in Eq. (3.21), and the mobility condition Eq. (3.23) can be rewritten as

$$a^{\ell+1} - 1 = (a-1)(1+pN) \quad p \in \mathbb{Z} \quad \Rightarrow \quad a^{\ell+1} = a \mod N. \tag{3.24}$$

Dividing both sides of the equation by $a$, we get that the mobility condition reads

$$a^{\ell} = 1 \mod M_a, \tag{3.25}$$

where $M_a \equiv N/\gcd(N,a)$. When $a$ and $N$ are coprime $M_a = N$ and, from Euler's totient theorem, there always exists a finite integer $\varphi(N)$ such that $a^{\varphi(N)} - 1 = 0 \pmod{N}$. In this case, Eq. (3.25) always admits a solution, $\ell = \varphi(N)$, and the connection to the order-finding problem is explicit (i.e. finding the smallest $r$ such that $a^r = 1 \mod N$). This concept raises an intriguing notion: even when dealing with exactly solvable Hamiltonians, there still exist some properties of the system, in this case, the emergent step size that particles can hop, that can be computationally expensive to determine for large enough $a$ and $N$. Finally, we note that the existence of solutions of Eq. (3.25) is a necessary but not sufficient condition for the existence of solutions of Eq. (3.23). In other words, while Eq. (3.25) might admit integer solutions $\ell$, the same is not necessarily true for Eq. (3.23). This follows from the fact that when isolating $a^{\ell+1}$ in Eq. (3.24) we threw some terms away, allowing emergent solutions $\ell$ that were not originally present in Eq. (3.23). Because of this, Euler's totient theorem does not ensure us that for any $a$ and $N$ coprime the mobility condition Eq. (3.23) can be satisfied.

Although unit charges and fluxes can be completely immobile, if Eq. (3.23) admits no solutions, single particles with higher charges (or flux) might be able to move. This follows from considering multiple powers, say $\beta$, of $M_{r_0,s}$ or $N_{r_0,s}$ operators on the vacuum. Following similar arguments as before, the condition for charges (or fluxes) of quantum number $\beta$ to move is the existence of an integer $\ell$ such that

$$\beta \, \alpha_\ell = 1 \mod N, \tag{3.26}$$

which is way less restrictive than the condition for unit charges and fluxes in Eq. (3.23). This is to be contrasted with 3D fracton physics, where higher charge configurations might be able to move ensuring they are arranged in a multi-particle configuration. Here, even single-particle configurations might move, if they carry appropriate quantum number $\beta > 1 \mod N$.

Now, a similar feature to usual 3D fractons is that isolated excitations are topologically robust and can be detected from far away by braiding mobile quadrupole excitations around them. One can explicitly check, for example, that when braiding the four-particles bound states associated to $W^{(1)}$ (see Fig. 7) on a closed curve $\gamma$, the wave function picks up a complex phase if there is a fractonic flux at $r_0$, contained inside. This follows from the fact that the $W^{(1)}(\gamma)$ and $N_{r_0,s}$ operators do not commute if $r_0$ is contained in $\gamma$. More generally, the complex phase, also known as mutual statistics between the two particles, is position dependent and is sensible to $r_0$.

The properties discussed above are very robust, characterizing the low-energy states as topologically ordered. In the presence of perturbations, say an external field

$$H \to H - g \sum_r X_r + \text{h.c.}, \tag{3.27}$$

the Hamiltonian is no longer exactly solvable. The low energy properties, however, survive ensured that the field is small enough $g \ll 1$. For a particle of unit charge/flux isolated in a disk region of radius $R$ (see Fig. 1), one needs to go in perturbation theory all the way up to order

$$\sum_{i=0}^{R} \alpha_i \geq R, \tag{3.28}$$

in order to see the isolated excitation hopping from its original position. In the relation above, we took into account that $\alpha_i \neq 1$ for all $i$, and at least one power of $X_r$ is necessary at each one of the $R$ sites $r$. We can also establish an upper bound for $\sum_{i=0}^{R} \alpha_i$ since $\alpha_i$ is defined mod $N$ and at max $N-1$ powers are needed in each site, and thus $\sum_{i=0}^{R} \alpha_i \leq (N-1)R$. Correspondingly, it takes a characteristic time of order

$$\tau \sim g^{-R}, \tag{3.29}$$

for the particle to tunnel from its position $r_0$ to any other point inside the disk. This is similar to a 3D Type-I fracton system, where a fracton can move one step at a time by emitting a dipole (which moves along a line until it is annihilated with another dipole emitted by another fracton), in order $\sim R$ in perturbation theory. The resulting characteristic time for a fracton to move $R$ steps in such a way and encounter another fracton is $\tau \sim R g^{-R}$.

## 4 Exponential Symmetries and Beyond

In this section, we discuss other models with even more exotic versions of exponential symmetries. In Sec. 4.1 we discuss the charge-flux attachment mechanism applied in the models previously studied in order to construct non-CSS codes. In Sec. 4.2 we discuss lattice models invariant under *dipolar exponential symmetries*, and finally in Sec. 4.3, we discuss the interplay of exponential and subsystem symmetries.

### 4.1 Non-CSS Codes: Chern-Simons-like Theories

At this stage, we have developed a taxonomy of generalized discrete gauge theories from *exponential polynomial symmetries*. These generalized electromagnetic theories can be treated as CSS stabilizer codes, with ground states obeying zero charge and flux conditions. In this section, our aim is to create non-CSS codes exhibiting similar exponential charge conservation. Intriguingly, these theories can be manifested as Chern-Simons theories, where charge and flux are bonded together.

To set the stage, we begin with a modified Gauss law that features charge structures similar to the ones in Eq. (3.1), but with a modification: unit charges are bounded to unit gauge fluxes. This flux-charge binding process has the potential to give rise to a fractonic Chern-Simons theory in contrast to a Higgsed Maxwell theory.

In conventional Maxwell-type gauge theories, the ground state manifold is defined by projecting the local Hilbert space onto the vanishing net charge and flux sectors. With

the flux-charge binding effect, the net charge condition will be automatically satisfied, provided the theory is flux-free. Additionally, any excitation electric charge would also contain gauge flux, and vice versa, leading to fractional statistics between charges (and fluxes).

To create a Chern-Simons-type coupling, we assign the charge density to be equal to the local flux,

$$q_r = D_1 E_1 + D_2 E_2 \overset{!}{=} B_r, \tag{4.1}$$

where $B_r$ is defined in Eq. (3.7). A sufficient solution to this equation is to impose a local projection between the two sets of $\mathbb{Z}_N$ Pauli operators $X, Z$ and $\overline{X}, \overline{Z}$, such that

$$\bar{Z} = X^a, \quad \bar{X} = Z^{\dagger a}, \quad Z = \bar{X}^{\dagger a}, \quad X = \bar{Z}^a \tag{4.2}$$

It is worth noticing that Eq. (4.2) renders a self-consistent solution only if $a^2 = 1$ (mod $N$). In this case, the local Hilbert space per site is reduced from $N^2$ to $N$, with only one set of Pauli operators per site. The Hamiltonian in Eq. (3.9) is reduced to,

$$H = -X_r^{a+1} Z_r^{\dagger a+1} Z_{r+e_x} Z_{r-e_x}^a X_{r+e_y}^{\dagger a} X_{r-e_y}^\dagger + \text{h.c.}, \tag{4.3}$$

which is depicted in Fig. 10.

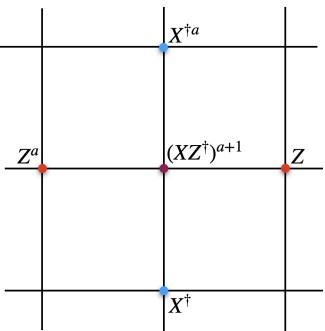

Figure 10: $\mathbb{Z}_N$ exponential symmetry operators under charge and flux attachment.

Now, there is only one stabilizer per site that projects both the charge and the flux to the null sector. As the constraint of flux and charge now becomes essentially the same thing, following the discussion in Sec. 3.3, it is not difficult to see that the ground state degeneracy is reduced to

$$\text{GSD}_{\text{CS}} = N \gcd(N_a, a^{L_x} - 1) \gcd(N_a, a^{L_y} - 1) \gcd(N_a, a^{L_x} - 1, a^{L_y} - 1). \tag{4.4}$$

This result is qualitatively different from the Higgsed Maxwell theory, where it was $\text{GSD}_{\text{CS}}^2$, as shown in Eq. (3.14).

It is worth noting that the solution given in Eq. (4.2) is not unique. Rather, it is a sufficient and straightforward solution that works when $a^2 = 1$ (mod $N$). It is possible that there are other solutions that satisfy the flux-charge binding constraint, but we will defer further investigation of them, as well as effective theories of such models, to future research.

## 4.2 Other Generalizations

We extend our discussion by introducing an alternative charge arrangement in Eq. (1.1), taking $g_r = 1, x, y, xy$ and allowing $f_r$ to take on the values of $x + y$. While we do not

plan to study the resulting theories in detail, we introduce the basic setup and general ideas, furnishing an example of an even more exotic exponential polynomial symmetry. The symmetry generators are explicitly given by

$$
\begin{aligned}
G &= \sum_r a^{x+y} q_r, \\
G_x &= \sum_r x\, a^{x+y} q_r, \\
G_y &= \sum_r y\, a^{x+y} q_r, \\
G_{xy} &= \sum_r xy\, a^{x+y} q_r.
\end{aligned}
\tag{4.5}
$$

The symmetries defined in Eq. (4.5) respect exponential charge, dipole, and quadrupole symmetries. The following Hamiltonian is a possible bosonic lattice realization of such conservation laws,

$$
b_r^{\dagger a^2} b_{r+e_x}^{2a} b_{r+2e_x}^{\dagger} + b_r^{\dagger a^2} b_{r+e_y}^{2a} b_{r+2e_y}^{\dagger} + \text{h.c.}.
\tag{4.6}
$$

The gauge structure, as before, can be studied by gauging the symmetries generated by Eq. (4.5). We couple the charged particles with a gauge potential $A_1, A_2$

$$
b_r^{\dagger a^2} b_{r+e_x}^{2a} e^{iA_{1,r+e_x}} b_{r+2e_x}^{\dagger} + b_r^{\dagger a^2} b_{r+e_y}^{2a} e^{iA_{2,r+e_y}} b_{r+2e_y}^{\dagger},
\tag{4.7}
$$

as illustrated as Fig. 11(a).

The $A_1$ and $A_2$ fields live on the sites of the square lattice as the conjugate partners of the electric field $E_1$ and $E_2$, respectively, and are subject to gauge transformations given by:

$$
\begin{aligned}
A_1 &\to A_1 + f_{r+e_x} + a^2 f_{r-e_x} - 2a\, f_r \\
A_2 &\to A_2 + f_{r+e_y} + a^2 f_{r-e_y} - 2a\, f_r
\end{aligned}
\tag{4.8}
$$

We hereby define the gauge invariant magnetic flux operator as

$$
\begin{aligned}
B_r &= a^2 A_{2,r-e_x} + A_{2,r+e_x} - 2a\, A_{2,r} \\
&\quad - a^2 A_{1,r-e_y} - A_{1,r+e_y} + 2a\, A_{1,r}.
\end{aligned}
\tag{4.9}
$$

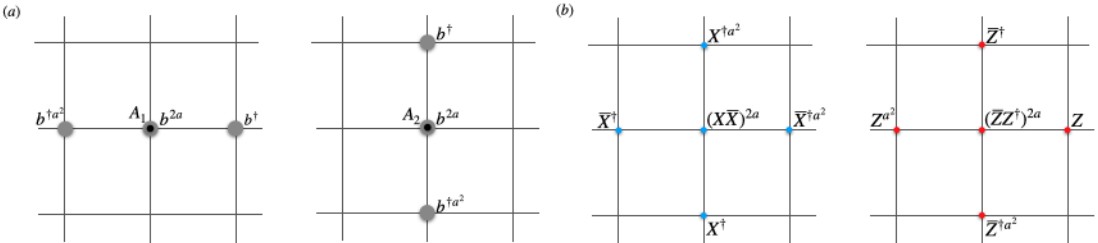

Figure 11: (a) Boson hopping structure, respecting all conserved quantities in Eq. (4.5); (b) $\mathbb{Z}_N$ charge and flux operators after gauging the exponential symmetries.

The gauge transformations and the gauge flux above define a $U(1)$ gauge theory, which likely to be confined in 2+1D. Since we are interested in deconfined phases we, again,

discretize the $U(1)$ gauge group down to $\mathbb{Z}_N$. There are two $\mathbb{Z}_N$ degrees of freedom at each vertex of the square lattice and the resulting gauge theory can be written as

$$H = -X_r^{2a}\, \bar{X}_r^{2a}\, \bar{X}_{r+e_x}^{\dagger a^2}\, \bar{X}_{r-e_x}^{\dagger}\, X_{r+e_y}^{\dagger a^2}\, X_{r-e_y}^{\dagger} - Z_r^{\dagger 2a}\, \bar{Z}_r^{2a}\, \bar{Z}_{r+e_y}^{\dagger}\, \bar{Z}_{r-e_y}^{\dagger a^2}\, Z_{r+e_x}\, Z_{r-e_x}^{a^2} + \text{h.c} \quad (4.10)$$

The low-energy physics of this model can be studied in a similar way as the one proposed in Sec. 3, which we do not detail here. In general, we expect it to behave similarly to the low-energy physics of Hamiltonian in Eq. (3.9), presenting an exotic manifestation of UV/IR mixing and constrained particle mobility. We expect these two models to differ from each other only in some minor aspects, as the explicit expression for the GSD and the particle-bound states configurations created at the endpoints of open string operators.

### 4.3 Landscape of Subsystem Exponential Symmetry: Spontaneous Symmetry Breaking, Symmetry Protected Topological Order and Fracton Topological Order

So far, we discussed the concept of exponential symmetries, characterized by charge operators with exponentially modulated spatial variation [20, 22]. Upon gauging these symmetries, the corresponding gauge theories demonstrate fracton behavior, notable for its exotic UV/IR mixing and constrained quasiparticle mobility.

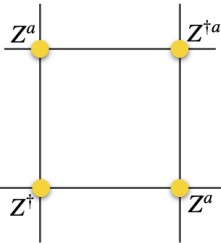

Figure 12: Subsystem exponential symmetry stabilizer $\mathbb{Z}_N$ model, defined on a square lattice.

Although we will not delve into a lot of details, in this section we introduce a novel type of symmetry, termed *subsystem exponential symmetry*. This symmetry is characterized by a charge operator with exponentially spatial modulation, conserved on sub-manifolds that encompass any x-rows and y-columns,

$$U^x(y) = \prod_i X_{r+ie_x}^{a^i}, \qquad U^y(x) = \prod_i X_{r+ie_y}^{a^i}. \quad (4.11)$$

The operators for such symmetry are reminiscent of subsystem symmetries, with the difference that the charge operator in each sub-manifold exhibits spatial modulation.

One can build a series of intriguing phases that exhibit various subsystem exponential symmetry patterns. A typical phase is the spontaneous symmetry-breaking phase leveraged by the following clock Hamiltonian on the 2D square lattice,

$$H = -Z_r^{\dagger}\, Z_{r+e_x}^{a}\, Z_{r+e_y}^{a}\, Z_{r+e_x+e_y}^{\dagger a^2} + \text{h.c.}. \quad (4.12)$$

The Hamiltonian terms are also shown in Fig. 12 and exhibit the symmetry refined in Eq. (4.11). A classical pattern of ground states of this Hamiltonian can be determined by arbitrarily choosing an orientation for the spins along a specific x-row and a specific

y-column. This suggests that the ground state entropy scales sub-extensively with the system size $L_x + L_y - 1$. The ground state pattern possesses long-range correlation, characterized by a non-vanishing four-point correlation function,

$$G(p, q) = \langle Z_r^\dagger \, Z_{r+qe_x}^{a^q} Z_{r+pe_y}^{a^p} Z_{r+qe_x+pe_y}^{\dagger a^{p+q}} \rangle. \tag{4.13}$$

Notably, this model reduces to the plaquette Ising Hamiltonian, explored in Ref. [49] for $a = 1$.

Symmetry and quantum entanglement come together and give rise to phases of matter nowadays known as Symmetry Protected Topological (SPT) phases. When the exponential subsystem symmetry we defined in Eq. (4.11) remains unbroken, the symmetry itself can still enrich quantum coherence, leading to a symmetry-protected topological order state with non-local quantum ordering and protected boundary modes. Here, we introduce a simple, exactly solvable Hamiltonian whose ground state achieves this state. Consider a decorated square lattice, where $\mathbb{Z}_N$ degrees of freedom lie at both vertices $r$ and plaquette centers $p$. We propose the following fixed-point Hamiltonian, also illustrated in Fig. 13

$$H = -Z_r^\dagger \, Z_{r+e_x}^a Z_{r+e_y}^a Z_{r+e_x+e_y}^{\dagger a^2} X_{r+\frac{e_x}{2}+\frac{e_y}{2}} - Z_p^{\dagger a^2} \, Z_{p+e_x}^a Z_{p+e_y}^a Z_{p+e_x+e_y}^\dagger X_{p+\frac{e_x}{2}+\frac{e_y}{2}} + \text{h.c.} \tag{4.14}$$

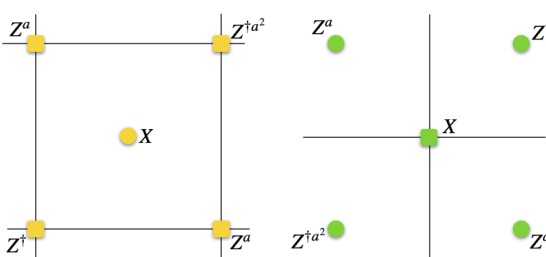

Figure 13: Operators defined on the decorated square lattice centered at plaquettes $p$ and vertices $r$, respectively.

The spins on the vertex site $r$ exhibit a set of exponential subsystem symmetries (denoted as $P^x$, $P^y$), and the same applies to the spins on the plaquette-centered sites $p$ (denoted as $R^x$, $R^y$),

$$P^x(y) = \prod_i X_{r+ie_x}^{a^i}, \qquad P^y(x) = \prod_i X_{r+ie_y}^{a^i},$$

$$R^x(y) = \prod_i X_{p+ie_x}^{a^{(L-i)}}, \qquad R^y(x) = \prod_i X_{p+ie_y}^{a^{(L-i)}}. \tag{4.15}$$

Here $L$ denotes the system size. Although we will not conduct a comprehensive study of this model, there are a few key points that signify its novelty. The Hamiltonian features a decorated defect structure, characterized by the frustrated plaquette Ising coupling between the four spins on the plaquette corners, decorated with a charge at the center. Consequently, the ground state displays non-vanishing membrane order

$$G(p, q) = \langle Z_r^\dagger Z_{r+qe_x}^{a^q} Z_{r+pe_y}^{a^p} Z_{r+qe_x+pe_y}^{\dagger a^{p+q}} \prod_{i=1}^q \prod_{j=1}^p X_{r+(i-1/2)e_x+(j-1/2)e_y}^{a^{i+j}} \rangle \tag{4.16}$$

The quantity $G(p, q)$ can be perceived as a non-local correlation where the four-point correlator among the spins located at the vertices is locked with the total exponential

charge of the spins residing at the center of the plaquette inside. Intriguingly, this non-vanishing membrane order implies that the SPT wave function, albeit can be prepared by a finite depth local unitary circuit, contains spurious long-range topological entanglement entropy, as proposed in Ref. [50].

Finally, we introduce an exotic fracton gauge theory in 3+1D, whose charge and flux excitations display subsystem (planar) exponential charge conservation. The model is situated on a cubic lattice, with degrees of freedom residing at the center $r$ of $xy$ plane plaquettes and at $z$ oriented edges $r'$, as indicated in Fig. 14. The proposed Hamiltonian for such anisotropic model is given by

$$
\begin{aligned}
H = &-Z_r^\dagger Z_{r+e_x}^a Z_{r+e_y}^a Z_{r+e_x+e_y}^{\dagger a^2} X_{r+e_z+\frac{e_x+e_y}{2}} X_{r-e_z+\frac{e_x+e_y}{2}}^\dagger \\
&- Z_{r'}^{\dagger a^2} Z_{r'+e_x}^a Z_{r'+e_y}^a Z_{r'+e_x+e_y}^\dagger X_{r'+e_z+\frac{e_x+e_y}{2}} X_{r'-e_z+\frac{e_x+e_y}{2}}^\dagger.
\end{aligned}
\tag{4.17}
$$

This Hamiltonian demonstrates a stabilizer structure that projects the local charge and flux into the vanishing sector. What distinguishes it from other stabilizer codes is that the charge and flux are not conserved globally. Instead, they display a unique exponential charge conservation on each $xz$ and $yz$ plane. We expect that mixing together both exponential and subsystem symmetries, even more exotic and rich fracton behaviors might emerge in 3+1D. With this, we hope our proposals of a landscape of subsystem exponential symmetries can inspire a broader vision of generalized symmetries and fracton gauge theories.

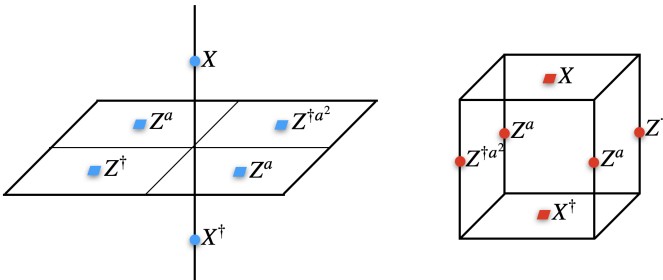

Figure 14: Operators defined on a 3D anisotropic cubic graph. The degrees of freedom sit both on the center of horizontal plaquettes (squares) and on vertical links (dots).

## 5 Discussion and Outlook

In this work, we present a class of $\mathbb{Z}_N$ gauge theories derived from gauging exponential symmetries. These gauge theories, specified in terms of stabilizer Hamiltonians, share many similarities with the toric code model. They exhibit topological ground state degeneracy and support anyonic excitations generated by non-local operators, while also displaying long-range entanglement beyond area law. However, they differ from the prominent toric code in prime aspects. Their charge and flux excitations have restricted mobility on the lattice, and their ground state degeneracies depend on parameters at the UV scale. In this sense, the exponential symmetry models resemble the 2D R2TC [13–15, 51], the 2D Dipolar Chern-Simons [25], and the X-Cube models [46, 52], but with some important differences. We conclude our discussion by comparing various aspects of these models in Table 1.

| Topological order | Symmetry generators | GSD | Immobile particles | $\tau$ |
|---|---|---|---|---|
| 2D Toric code | charge | finite | NO | $\mathcal{O}(g^{-1})$ |
| 2D Rank-2 Toric code | vector charge | finite | NO | $\mathcal{O}(g^{-N^2})$ |
| 2D Dipolar CS | charge, dipole and quadrupole | finite | NO | $\mathcal{O}(g^{-N^2})$ |
| Models in this paper | exponential charges | finite | YES | $\mathcal{O}(g^{-R})$ |
| 3D X-Cube et.al. | subsystem charges | sub-extensive | YES | $\mathcal{O}(R\,g^{-R})$ |

Table 1: Comparison of various aspects of 2D and 3D discrete gauge theories. The tunneling time $\tau$ is estimated in perturbation theory by adding a small external field $g \ll 1$.

The models discussed in the previous paragraph can be obtained by gauging unusual global symmetries, as elucidated in Ref. [53], and specified in Table 1. The table also summarizes the presence, or not, of immobile particles in the theory spectrum, as well as the scaling of the ground-state degeneracy with system size. Since the spatially modulated $\mathbb{Z}_N$ symmetries are only well-defined when the system sizes are compatible, the resulting ground state degeneracies for the two-dimensional R2TC, Dipolar Chern-Simons, and the exponential symmetry models all depend on the system size. The 3D fracton codes, as X-cube [46] and checkerboard codes [54], however, exhibit ground state degeneracies that grow with the system length due to the sub-extensive number of independent conservation laws generated by subsystem symmetries. We also present, in the table, the characteristic tunneling time for a single isolated quasiparticle in a region of radius $R$ (see Fig. 1) to hop from its position under an external field perturbation $g \ll 1$.

We would like to note that there exist richer classes of other fracton models (mostly in 3D) that are beyond the summary in Table 1. These models showcase even richer content that transcends the generalized gauge symmetry perspective, including includes twisted fracton gauge theories, cage-net models, hybrid fracton phases, and non-Abelian fracton models.

We hope that the stabilizer codes and discrete gauge theories proposed in this paper might open a new chapter in the exploration of generalized symmetries and fracton gauge theories. Here, we highlight several open questions that warrant further investigation in the future: 1) While the majority of topological phase transitions between deconfined gauge theories and Higgsed (or confined) phases have been explored in the early literature, it would be compelling to investigate possible phase transitions by adding onsite transverse fields that could potentially trigger a transition between the deconfined phase displayed in this manuscript and a confined phase. 2) Inspired by the double-semion model as a twisted $\mathbb{Z}_2$ gauge theory in 2+1D, we anticipate that one could develop a similar twisted gauge theory with exponential charge conservation. Another relevant direction would be to extend the $\mathbb{Z}_N$ gauge theory explored in this manuscript into U(1) and seek possible deconfined U(1) phases by adding Chern-Simons-like couplings. 3) As highlighted in Sec.2, exponential symmetries cannot be considered internal symmetries since their generators do not commute with spatial translations. As a result, spatial symmetry defects, such as dislocations, can permute between different topological charge sectors, making it engrossing to investigate the effects of lattice defects as they could potentially trap charged zero modes.

*Note: During the completion of this paper, we learned of an independent work currently in preparation that develops the 3D generalization of gauging exponential symmetries [55].*

# Acknowledgements

We are grateful to Meng Cheng, Haruki Watanabe, Heitor Casasola, and Gustavo Yoshitome for their helpful feedback and valuable discussions. This work was completed in part at Aspen Center for Physics (Y.Y.), which is supported by National Science Foundation grant PHY-2210452 and Durand Fund. This work is also supported by the DOE Grant No. DE-FG02-06ER46316 (G. D. and C.C.).

# Appendix

## 5.1 Ground State Degeneracy and Global Constraints in Exponential Charge Gauge Theory

In this appendix, we derive the ground state degeneracy in Eq. (3.14) for the exponential charge model in Eq. (3.9). First, let us note there are two $\mathbb{Z}_N$ degrees of freedom per site, giving rise to a total Hilbert space of dimension $\dim \mathcal{H} = N^{2N_{\text{sites}}}$, where $N_{\text{sites}}$ is the number of sites on the lattice. There are, as well, two $\mathbb{Z}_N$ operators with $N$ distinct eigenvalues $\exp 2\pi q_r\, N$ - the charge $\mathcal{Q}_r$ and the flux $\mathcal{B}_r$ - per site, being able to distinct and label $N^{2N_{\text{sites}}}$ states. The degeneracy, however, comes from the fact that under periodic boundary conditions, not all the eigenvalues $q_r = 0, \ldots, N-1$ for different lattice sites are independent. Instead, they are subject to global constraints, as in Eq. (3.12), which here we carefully derive.

The $\rho$ factors are extremely important, telling us how many distinct states are actually labeled by the same $\mathcal{Q}_r$ and $\mathcal{B}_r$ eigenvalues. We note that the flux $\mathcal{B}_r$ operators also obey the same equations as $\mathcal{Q}_r$ in Eq. (3.12) and that each one of the constraints contributes to the ground state degeneracy. The role of each $\rho$ factor is to "reduce" the $\mathbb{Z}_N$ operators in the products (3.12) down to a $\mathbb{Z}_{N_a/\rho}$ ones. The ground state degeneracy is given by the number of such different configurations $(N \times N_a/\rho_1 \times N_a/\rho_2 \times N_a/\rho_{12})^2$, simplifying to

$$\dim \mathcal{H}_0 = \left[ N \gcd(N_a, a^{L_x} - 1) \gcd(N_a, a^{L_y} - 1) \gcd(N_a, a^{L_x} - 1, a^{L_y} - 1) \right]^2, \quad (5.1)$$

where the power of 2 takes into account the constraints for both the charge and flux operators.

The way we count the global constraints in Eq. (3.12) is to find the sets of solutions $t_r \mod N$ to the product of charge operators in different sites $r = (x, y)$

$$\prod_r \mathcal{Q}_r^{t_r} = \mathbb{1}. \quad (5.2)$$

We inherit the discussion present in Ref. [22] based on Euler's totient theorem; we consider the case when $a$ and $N$ are not necessarily coprime but $a \neq 0 \mod \text{rad}(N)$. In such a case, one can always define $N_a$ as the greatest divider of $N$ that is coprime to $a$. From Euler's theorem, there always exists a finite integer $\varphi(N_a)$ such that $a^{\varphi(N_a)} - 1 = 0 \,(\text{mod } N_a)$ with $\varphi(n)$ being the Euler's totient function, allowing us to find the $\rho$ factors.

In the following we show that the possible $t_r$ solutions are $t_r = t, \rho_1 a^x, \rho_2 a^y, \rho_{12} a^{x+y}$, for integers $t, \rho_1, \rho_2, \rho_{12}$. The functional form of these solutions can be traced back to the conservation laws of usual charge and exponential charges in Eq. (3.1). To see this, we use the explicit form of $\mathcal{Q}_r$ and rewrite Eq. (5.2) in terms of Pauli operators coming from different $A_r$ acting on a single site $r$

$$\prod_r (X_r)^{(a+1)t_r - at_{r-e_y} - t_{r+e_y}} \left( \bar{X}_r \right)^{(a+1)t_r - t_{r+e_x} - at_{r-e_x}} = \mathbb{1}.$$

We thus see that the only way for the constraint (5.2) to hold globally is that each term in the product above reduces to the identity. The problem then translates into solving a pair of recursive modular equations

$$
\begin{aligned}
(a+1)t_r - at_{r-e_y} - t_{r+e_y} &= 0 \mod N, \\
(a+1)t_r - t_{r+e_x} - at_{r-e_x} &= 0 \mod N,
\end{aligned}
\tag{5.3}
$$

which we solve, in some detail. The ground state degeneracy, in the end, boils down to counting how many independent solutions such equations admit.

First, let us note that the constant solution $t_r = t$, with $t$ an integer mod $N$ satisfies the two equations above, providing us with $N$ independent solutions. This is the $N$ factor in the GSD expression in Eq. (3.14). Another class of solutions, which is less trivial than the constant one is the exponential

$$
t_r = \rho_1 \, a^x \mod N
\tag{5.4}
$$

where $x$ is the horizontal coordinates for the lattice point $r$. As mentioned in the main text, we consider only $a \neq 0 \mod \mathrm{rad}(N)$ such that $a^x$ never vanishes mod $N$. Under periodic boundary conditions, the values $\rho_1$ must obey

$$
\begin{aligned}
\rho_1 a^x &= \rho_1 a^{x+L_x} \mod N \quad \forall \, x = 1, \dots, L_x, \\
&\Rightarrow \rho_1 \left( a^{L_x} - 1 \right) = 0 \mod N_a
\end{aligned}
\tag{5.5}
$$

whose solutions are

$$
\rho_1 = \frac{N_a}{\gcd(N_a, a^{L_x} - 1)} k,
\tag{5.6}
$$

where $k$ is an integer number mod $\gcd(N_a, a^{L_x} - 1)$. Thus, the ansatz in Eq. (5.4) assumes $\gcd(N_a, a^{L_x} - 1)$ distinct solutions, contributing with this number to the ground state degeneracy expression in Eq. (3.14). From similar arguments, the class of solutions

$$
t_r = \rho_2 \, a^y \mod N
\tag{5.7}
$$

contributes with a factor of $\gcd(N_a, a^{L_y} - 1)$ to GSD. Finally,

$$
t_r = \rho_{12} \, a^{x+y},
\tag{5.8}
$$

are also solutions to the recursive equations (5.3). Here, the periodic boundary conditions require simultaneously that

$$
\rho_{12} \left( a^{L_x} - 1 \right) = 0 \mod \quad \text{and} \quad \rho_{12} \left( a^{L_y} - 1 \right) = 0 \mod N_a,
\tag{5.9}
$$

which can be solved by

$$
\rho_{12} = \frac{N_a}{\gcd(N_a, a^{L_x} - 1, a^{L_y} - 1)} p,
\tag{5.10}
$$

with $p$ an integer mod $\gcd(N_a, a^{L_x} - 1, a^{L_y} - 1)$. The integer $p$ parameterizes how many independent solutions the ansatz in Eq. (5.7) assumes, contributing with a factor of $\gcd(N_a, a^{L_x} - 1, a^{L_y} - 1)$ to the expression for the GSD.

Using similar arguments, we can also show that the flux operators $\mathcal{B}_r$ are subject to global constraints

$$
\prod_r \mathcal{B}_r^{\tilde{t}_r} = \mathbb{1},
\tag{5.11}
$$

as well. From inspection, it is straightforward to show that $\tilde{t}_r$ obeys similar equations as $t_r$ in (5.3) and, consequently, obeys similar solutions as well. When taking these into account, the ground state degeneracy in Eq. (3.14) acquires an overall power of 2.

## 5.2 Wilson Lines in Exponential Charge Gauge Theory

We now apply this protocol to our model, proposed in Sec. 3. Upon Higgsing, we used an exponential map from $A_{i,r}$ and $E_{i,r}$ to the $\mathbb{Z}_N$ operators $Z$ and $X$. It means that the sums (integrals) in usual gauge theories translate into products in the $\mathbb{Z}_N$ model. The ground state is projected to zero local fluxes with eigenvalue $\mathcal{B}_r = 1$ for all lattice sites $r$. Thus, the total flux on any open or closed area is subject to $\prod_r \mathcal{B}_r = 1$. In addition, one can also derive that the eigenvalues of the exponential fluxes also multiply up to one, $\mathbb{Z}_N$ analogs of the fluxes in Eq. (3.8).

Consider now the ground state wave function on an open cylinder, as shown in Fig. 6. We focus on the Wilson lines that are defined along the y-loops, associated to the first two conservation laws in Eq. (3.8) yielding,

$$\prod_{r \in \mathcal{A}} \mathcal{B}_r = \prod_i Z^a(0, i) \, Z^\dagger(1, i) \prod_j Z^{\dagger a}(x_0 - 1, j) \, Z(x_0, j),$$
$$\prod_{r \in \mathcal{A}} \mathcal{B}_r^{a^{x_0 - x}} = \prod_i Z^{a^{x_0}}(0, i) \, Z^{-a^{x_0}}(1, i) \prod_j Z^\dagger(x_0, j) \, Z(x_0, j). \tag{5.12}$$

Based on Eq. (5.12), the total flux on an open cylinder is reduced to the holonomy operators localized at the boundaries,

$$W^{(1)}(x) = \prod_i Z^a(x, i) \, Z^\dagger(x + 1, i), \quad \text{and} \quad W^{(2)}(x) = \prod_i Z(x, i) \, Z^\dagger(x + 1, i). \tag{5.13}$$

It follows from the flux conservation law that these two operators are constrained to obey

$$W^{(1)}(x) \, W^{(1)\dagger}(x + 1) = 1,$$
$$W^{(2)\dagger a}(x - 1) \, W^{(2)}(x) = 1. \tag{5.14}$$

Following the same procedure for the last two conservations laws in Eq. (3.8), when integrated on a cylinder $\mathcal{A}$ as in Fig. 6, they decompose into products

$$\prod_{r \in \mathcal{A}} \mathcal{B}_r^{a^{L_y - y}} = \prod_i Z^{a^{L_y - i + 1}}(0, i) \, Z^{\dagger a^{L_y - i}}(1, i) \prod_j Z^{\dagger a^{L_y - j + 1}}(x_0 - 1, j) \, Z^{a^{L_y - j}}(x_0, j),$$
$$\prod_{r \in \mathcal{A}} \mathcal{B}_r^{a^{x_0 - x} a^{L_y - y}} = \prod_i Z^{a^{L_y - i + x_0}}(0, i) \, Z^{\dagger a^{L_y - i + x_0}}(1, i) \prod_j Z^{\dagger a^{L_y - j}}(x_0, j) \, Z^{a^{L_y - j}}(x_0, j). \tag{5.15}$$

These engender another set of Wilson line operators

$$W^{(3)}(x) = \prod_i Z^{a^{L_y - i + 1}}(x, i) \, Z^{\dagger a^{L_y - i}}(x + 1, i),$$
$$\text{and} \quad W^{(4)}(x) = \prod_i Z^{a^{L_y - i}}(x, i) \, Z^{\dagger a^{L_y - i}}(x + 1, i). \tag{5.16}$$

Following the flux conservation laws, these operators obey

$$W^{(3)}(x) W^{(3)\dagger}(x + 1) = 1,$$
$$W^{(4)\dagger a}(x - 1) W^{(4)}(x) = 1. \tag{5.17}$$

Similarly, one can define the dual operators going along the $x$-direction. We characterize the charge sectors by integrating the conserved quantities in Eq. (3.1) on an open cylinder

along the $y-$ direction (a rotated cylinder when compared to the one in Fig. 6.,

$$V_x^{(1)}(y) = \prod_i X^\dagger(i,y)\, X^a(i, y+1), \qquad V_x^{(3)}(y) = \prod_i X^{\dagger a^i}(i,y)\, X^{a^{i+1}}(i, y+1),$$

$$V_x^{(2)}(y) = \prod_i X^\dagger(i,y)\, X(i, y+1), \qquad V_x^{(4)}(y) = \prod_i X^{\dagger a^i}(i,y)\, X^{a^i}(i, y+1), \quad (5.18)$$

Likewise, the Wilson lines composed of $\bar{X}, \bar{Z}$ also emerge at the boundary of cylinder regions,

$$W_x^{(1)}(y) = \prod_i \bar{Z}^a(i,y)\, \bar{Z}^\dagger(i, y+1), \qquad V_y^{(1)}(x) = \prod_i \bar{X}^\dagger(x,i)\, \bar{X}^a(x+1, i),$$

$$W_x^{(2)}(y) = \prod_i \bar{Z}(i,y)\, \bar{Z}^\dagger(i, y+1), \qquad V_y^{(2)}(x) = \prod_i \bar{X}^\dagger(x,i)\, \bar{X}(x+1, i),$$

$$W_x^{(3)}(y) = \prod_i \bar{Z}^{a^{L_x-i+1}}(i,y)\, \bar{Z}^{\dagger a^{L_x-i}}(i, y+1), \qquad V_y^{(3)}(x) = \prod_i \bar{X}^{\dagger a^i}(x,i)\, \bar{X}^{a^{i+1}}(x+1, i)$$

$$W_x^{(4)}(y) = \prod_i \bar{Z}^{a^{L_x-i}}(i,y)\, \bar{Z}^{\dagger a^{L_x-i}}(i, y+1), \qquad V_y^{(4)}(x) = \prod_i \bar{X}^{\dagger a^i}(x,i)\, \bar{X}^{a^i}(x+1, i),$$

where all these $V^{(i)}$ and $W^{(i)}$ obey similar conditions to Eq. (5.14) and (5.17).

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
