# Peer review of "D Fractons from Gauging Exponential Symmetries"

_SciPost Physics_

## Round 3 · Referee Report · Anonymous (Referee 1) · 2023-11-9

Report

In this work, the authors make an argument that there are the 2D fracton topological orders in the two-dimensions. However, the previous work for a long time in the community has shown from various ways that fracton topological orders can only exist in 3D and higher. This obvious conflict must be addressed explicitly and clearly in the main text. The authors should explain why their conclusions are different from D. Aasen, D. Bulmash, A. Prem, K. Slagle and D. J. Williamson, Phys. Rev. Research 2, 043165 (2020) and J. Haah, arXiv:1812.11193. The former paper is cited as Ref. [43] in this manuscript and only appears once (page 3) and latter paper is not cited. There are detailed discussions on the issues in Section V in Ref. [43]. I think the authors must explain why their models can avoid the no-go result given by Ref. 43 and also compare their results with Haah's analysis. Unfortunately, I did not see such discussions in the present manuscript. In fact, the quantity R can be very large but always finite, which means that the excitations can be moved at the end of day. We must agree that, defining a phase must be in the thermodynamical limit.

In addition, a technical concern is that, in general, we should perform symmetry transformation explicitly on the field (boson/electron creation/annihliation) operators to demonstrate how operators are transformed under the symmetry operation. After this is clearly done, one can safely do the gauging by Peierls substitution. In Section 2.1, I did not see such standard procedure but a direct shift into gauging shown in eqs. 2.4. I think it is important to perform the above standard procedure carefully as gauging a group must be done after the symmetry operation is clearly defined.

---

## Round 3 · Referee Report · Anonymous (Referee 2) · 2024-9-7

Report

In this paper, the authors studied systems with a so-called exponential symmetry. The authors gauged the symmetry in 2+1D systems and found topological / fracton type of behavior in the resulting gauge theory.

One highly surprising result is that the authors claim there are 2D stabilizer models with a finite ground state degeneracy, but contain immobile quasi-particles (see for example Table I). This result is against common expectation that 2D stabilizer models cannot contain immobile quasi-particle (that fracton behavior is a 3D behavior) and finite ground state degeneracy indicates topological order (with only mobile quasi-particles). The authors mentioned an example of this type in section 3. The authors claim that the interesting cases can show up when a and N share common factors under prime decomposition. But then it is not clear how the condition a^L-1 = 0 mod N can be satisfied which is needed to be compatible with periodic boundary condition. This example is so surprising that I highly recommend the authors explore the model in more depth and explain how the properties mentioned above are possible.

I cannot determine the scientific value of this work before more in depth analysis is given. The result is either completely unexpected and groundbreaking or not correct/consistent.

Recommendation

Ask for major revision

---

## Editorial Decision

awaiting_resubmission